# AMORTISING THE GAP BETWEEN PRE-TRAINING AND FINE-TUNING FOR VIDEO INSTANCE SEGMENTATION

## ABSTRACT

Video Instance Segmentation (VIS) development heavily relies on fine-tuning pre-trained models initially trained on images. However, there is often a significant gap between the pre-training on images and fine-tuning for video, which needs to be noticed. In order to effectively bridge this gap, we present a novel approach known as *"video pre-training"* to achieve substantial improvements in VIS. Notably, our approach has enhanced performance on complex video datasets involving intricate instance relationships. Our primary contribution is minimizing disparities between the pre-training and fine-tuning stages at both the data and modeling levels. Specifically, we introduce the concept of consistent pseudo-video augmentations to enrich data diversity while maintaining instance prediction consistency across both stages. Additionally, at the modeling level for pre-training, we incorporate multi-scale temporal modules to enhance the model's understanding of temporal aspects, allowing it to better adapt to object variations and facilitate contextual integration. One of the strengths of our approach is its flexibility, as it can be seamlessly integrated into various segmentation methods, consistently delivering performance improvements. Across prominent VIS benchmarks, our method consistently outperforms all state-of-the-art methods. For instance, when using a ResNet-50 as a backbone, our approach achieves a remarkable 4.0% increase in average precision (AP) on the most challenging VIS benchmark, OVIS, setting a new record. The code will be made available soon.

## 1 INTRODUCTION

Video Instance Segmentation (VIS) is an integrated task encompassing the concurrent processes of classifying, segmenting, and tracking instances within a video sequence. Since first introduced by Yang et al. (2019a), this task has facilitated a wide range of applications, including but not limited to autonomous driving, video editing, and video understanding. Existing VIS methodologies are typically categorized into two primary paradigms: online (Yang et al., 2019a; 2021; Huang et al., 2022; Wu et al., 2022b; Heo et al., 2022a; Ying et al., 2023; Zhang et al., 2023) and offline (Wang et al., 2021; Cheng et al., 2021a; Wu et al., 2022a; Heo et al., 2022b; Zhang et al., 2023) methods. Online VIS methodologies perform frame-by-frame segmentation of video sequences and then utilize matching techniques to track instances. In contrast, offline VIS approaches process the entire video sequence in a single pass, seamlessly integrating tracking and segmentation. While both online and offline methods offer unique advantages, they share a fundamental reliance on a robust pre-trained model for image-level instance segmentation as their foundation.

While current methods have undoubtedly pushed the boundaries of VIS, models primarily pre-trained on images introduce limitations to the performance of VIS, especially when dealing with lengthy and intricate video sequences. This limitation stems primarily from the more significant cost and complexity associated with annotating video data compared to images, resulting in relatively small video datasets that often consist of just a few hundred or thousand videos. Consequently, contemporary VIS research has been compelled to rely on image pre-training. However, to bridge the divide between images and videos, existing VIS approaches require the integration of increasingly complex temporal modules to furnish richer information for tracking instances within video sequences. Heo et al. (2022b); Yang et al. (2021); Hwang et al. (2021) have explored leveraging pre-trained models with parameters initialized toward video tasks. However, the frame count during pre-training is typically set to 1, which aligns with image pre-training paradigms. Consequently, these methods

Figure 1: Pipelines of previous methods and our method. The difference lies in that we utilize the pre-trained temporal model rather than image model as the initialized weights, which improves the consistency of the pre-trained and fine-tune stages.

yield more significant improvements when applied to more straightforward video datasets, with their fundamental enhancements rooted in image segmentation.

Given the scarcity of video data, several studies Ying et al. (2023); Zhang et al. (2023); Heo et al. (2022a); Wu et al. (2022b); Heo et al. (2022b) have put forth methods that rely on data augmentation to create pseudo-videos as a means to tackle this limitation. Nonetheless, these methods frequently neglect the critical aspect of preserving consistency between successive frames during the pseudo-video generation process. These data augmentation techniques, including Cutmix Yun et al. (2019b); French et al. (2019); Yun et al. (2019a), involve the random selection of an image from the dataset and the subsequent pasting of a portion of it onto images or videos. However, maintaining temporal coherence between frames is often overlooked in the application of these techniques. While this approach can be effective in improving dataset diversity for image segmentation tasks, it can pose challenges in the context of complex video tasks, especially when dealing with numerous instances that exhibit similarities in categories, sizes, and shapes. Consequently, we contend that there is a pressing need for a more refined methodology that not only enriches data diversity but also guarantees the consistency of video sequences.

Many online VIS(Yang et al., 2019a; 2021; Huang et al., 2022; Wu et al., 2022b; Heo et al., 2022a; Ying et al., 2023; Zhang et al., 2023) methods typically carry out association and tracking on the video sequence after completing single-frame segmentation to maintain segmentation performance. However, our results demonstrate that engaging with the video sequence through temporal modules during the segmentation phase can be significantly beneficial in enhancing performance. Nevertheless, exceptions exist. Offline VIS(Wang et al., 2021; Cheng et al., 2021a; Wu et al., 2022a; Heo et al., 2022b; Zhang et al., 2023) methods often introduce an excessive number of temporal modules during fine-tuning, which can lead to subpar single-frame segmentation performance. In response to this challenge, the DVIS approachZhang et al. (2023) employed a somewhat cumbersome strategy. It initially froze the pre-trained model to train an online model and then froze all parameters of the online model, subsequently training a new set of temporal modules for the offline method. Consequently, we propose a more efficient approach: directly incorporating temporal modules during pre-training with video inputs.

Given the scarcity of video data, several studies Ying et al. (2023); Zhang et al. (2023); Heo et al. (2022a); Wu et al. (2022b); Heo et al. (2022b) have put forth methods that rely on data augmentation to create pseudo-videos to tackle this limitation. Nonetheless, these methods frequently need to pay more attention to preserving consistency between successive frames during the pseudo-video generation process. These data augmentation techniques, including Cutmix Yun et al. (2019b); French et al. (2019); Yun et al. (2019a), involve randomly selecting an image from the dataset and pasting a portion onto images or videos. However, maintaining temporal coherence between frames is often overlooked in applying these techniques. While this approach can effectively improve dataset diversity for image segmentation tasks, it can pose challenges in complex video tasks, especially when dealing with numerous instances that exhibit similarities in categories, sizes, and shapes. Consequently, there is a pressing need for a more refined methodology that enriches data diversity and guarantees video sequences' consistency.

Many online VIS(Yang et al., 2019a; 2021; Huang et al., 2022; Wu et al., 2022b; Heo et al., 2022a; Ying et al., 2023; Zhang et al., 2023) methods typically carry out association and tracking on the video sequence after completing single-frame segmentation to maintain segmentation performance. However, our results demonstrate that engaging with the video sequence through temporal modules during the segmentation phase can significantly enhance performance. Nevertheless, exceptions exist. Offline VIS(Wang et al., 2021; Cheng et al., 2021a; Wu et al., 2022a; Heo et al., 2022b; Zhang et al., 2023) methods often introduce excessive temporal modules during fine-tuning, leading to subpar single-frame segmentation performance. In response to this challenge, the DVIS approachZhang et al. (2023) employed a somewhat cumbersome strategy. It initially froze the pre-trained model to train an online model and then froze all parameters of the online model, subsequently training a new set of temporal modules for the offline method. Consequently, we propose a more efficient approach: directly incorporating temporal modules during pre-training with video inputs.

In this paper, we argue that the primary bottleneck in VIS stems from the disparities between the pre-training and fine-tuning stages, encompassing data and model aspects. To tackle these challenges, we present a novel video pre-training method. Specifically, we introduce a consistent pseudo-video augmentation approach at the data level to maximize the potential for maintaining consistency within the generated pseudo-video sequences. In generating pseudo videos, we employ a technique where we replicate the same image multiple times, with the number of replications determined by the desired length of the pseudo video. Subsequently, we select an instance from the current frame and, to preserve video consistency, subject it to a series of subtle and random augmentations before pasting it into every frame of the pseudo videos. This approach effectively replicates instance motion and addresses common occlusion scenarios commonly encountered in videos. It proves particularly valuable when dealing with complex video datasets where instances often share strikingly similar sizes, shapes, and colors, making their distinction challenging. At the modeling level, we propose the integration of a multi-scale temporal module that encompasses both short-term and long-term temporal considerations. This module is utilized in both the pre-training and fine-tuning phases, and its addition is strategically aimed at enhancing the model's capacity during pre-training. It achieves this by improving the model's comprehension of temporal dynamics across various time scales and facilitating the contextualization of information. Initializing these temporal modules during pre-training is paramount for success in downstream video-related tasks. In summary, our approach offers several noteworthy contributions:

- We introduce an innovative video pre-training method that utilizes consistent pseudo-video augmentation techniques to enrich the diversity of pseudo-videos while preserving their consistency. This augmentation strategy effectively narrows the gap between pre-training and video fine-tuning at the data level, resulting in improved performance for both image and video tasks.

- To infuse temporal knowledge into the pre-trained temporal model, our method incorporates a straightforward yet highly effective multi-scale temporal module during pre-training. This module encompasses short-term and long-term temporal considerations, aiming to bolster the model's capabilities through pre-training and video fine-tuning. It achieves this by enhancing the model's understanding of temporal dynamics across different time scales and facilitating the contextualization of information.

- Our approach establishes new benchmarks in Video Instance Segmentation (VIS), achieving state-of-the-art performance across three major benchmarks: OVIS, YouTube-VIS 2019, and 2021. Furthermore, the modules we introduce can seamlessly integrate into other segmentation or detection methods, offering substantial performance improvements.

## 2 RELATED WORK

### 2.1 INSTANCE SEGMENTATION

In the realm of instance segmentation, researchers have explored two predominant categories of approaches: traditional instance segmentation methods (He et al., 2017; Ren et al., 2017; Chen et al., 2019; Long et al., 2015) and transformer-based techniques (Cheng et al., 2022; Carion et al., 2020; Zhu et al., 2020; Zhang et al., 2021; Fang et al., 2021; Li et al.; Zhang et al.). These approaches have proven effective in predicting binary masks associated with object instances and their corresponding class labels. Traditional instance segmentation methods have historically been rooted in object

detection models. Their objective is to predict a set of binary masks and corresponding categories. For instance, Mask R-CNN (He et al., 2017) extends the Faster R-CNN (Ren et al., 2017) detection model by incorporating a parallel mask branch alongside the detection branch. While it has significantly advanced instance segmentation, it relies on numerous heuristic designs and the careful tuning of multiple hyperparameters.

The prevailing trend centers around fine-tuning primarily on Transformer-based pre-training models specialized for instance segmentation. Transformer-based methods have substantially simplified the instance segmentation pipeline and have gained widespread recognition in the VIS domain. These models harness the capabilities of Transformers, initially designed for natural language processing, and adapt them to computer vision. One notable innovation in this regard is the Mask2Former approach (Cheng et al., 2022), which introduces masked attention mechanisms into the Transformer architecture. This approach unifies various segmentation tasks, including instance, semantic, and panoptic segmentation (Cheng et al., 2021b; Zhang et al., 2021). These Transformer-based methods represent a significant leap forward in VIS, showcasing the adaptability of Transformer architectures to complex computer vision tasks.

## 2.2 VIDEO INSTANCE SEGMENTATION

**Online VIS methods** operate in real-time and often rely on image-level instance segmentation models as their foundation. MaskTrack R-CNN(Yang et al., 2019a), for instance, extends the Mask R-CNN(He et al., 2017) architecture by introducing a tracking head for associating instances across video frames using heuristic cues. IDOL(Wu et al., 2022b) employs a memory bank during inference to match newly detected foreground instance embeddings with previously stored embeddings from prior frames. Another noteworthy approach is CTVIS(Ying et al., 2023), which draws inspiration from online methods' inference stage to learn more robust and discriminative instance embeddings during training. Most existing Online VIS methods introduce temporal modules after the segmentation is completed. Typically, segmentation and tracking are treated as separate entities. However, introducing simple temporal modules during the segmentation phase can enhance tracking capabilities without compromising segmentation performance. Further, augmentation through video pre-training to initialize these temporal modules is anticipated to yield even more significant enhancements.

**Offline VIS methods** process the entire video in a single pass, using the entire video context during inference. Mask2Former-VIS(Cheng et al., 2021a) and SeqFormer(Wu et al., 2022a) leverage attention mechanisms to process spatio-temporal features and directly predict instance mask sequences. VITA(Heo et al., 2022b) proposes decoding video object queries from sparse frame-level object tokens rather than dense spatio-temporal features to address memory constraints on highly long videos.DVIS(Zhang et al., 2023) is a two-stage approach. Initially, DVIS freezes the parameters of Mask2Former and introduces a referring tracker module, thereby realizing an online model. Subsequently, the parameters of the online model are frozen, and a temporal refiner is added to achieve an offline model. The current trend in offline VIS methods is characterized by an overabundance of temporal modules introduced during the segmentation process, leading to subpar single-frame segmentation performance. In order to address this issue, we contend that a more efficient approach is to seamlessly integrate multi-scale temporal modules with consistent pseudo-video augmentations at the pre-training phase.

## 2.3 DATA AUGMENTATION

Currently, the primary focus of research lies in the realm of image augmentation(Zhao et al., 2022; Cubuk et al.; Yun et al., 2019b; French et al., 2019; Yun et al., 2019a), with a notable scarcity of methodologies explicitly tailored for augmenting videos. AugSeg (Zhao et al., 2022) is an example of an approach that employs an auto-augmentation strategy (Zhao et al., 2022; Cubuk et al., 2020; Cubuk et al.), where random data transformations are selected with distortion intensities uniformly sampled from a continuous space. In the context of VIS, many existing methodologies rely on data augmentation techniques to generate pseudo-video sequences. However, these methods often resort to basic rotations. We introduce a novel augmentation method to enhance the VIS task, as detailed in Tab. 7 and Tab. 4. Our findings demonstrate that this innovative augmentation technique significantly improves instance segmentation and VIS performance.

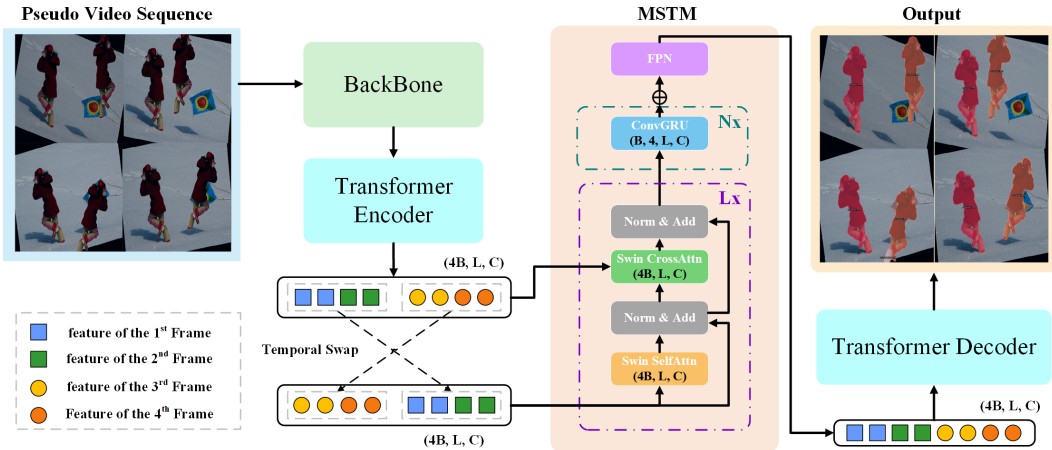

Figure 2: Detailed pipeline of our method. The pseudo and training videos are inputted to the VIS temporal network. The temporal network is equipped with a multi-scale temporal block. MSTM: multi-scale temporal module.

# 3 METHOD

The typical line of VIS methods involves pre-training instance segmentation models on image datasets and then fine-tuning the models with additional temporal modules on video datasets. As mentioned above, there is a significant gap between the per-training stage and fine-tuning

Table 1: Augmentation pool used in *pseudo video*.

| Identity | Autocontrast | Equalize | Gaussian blur |
|----------|--------------|----------|---------------|
| Contrast | Sharpness | Color | Brightness |
| Hue | Posterize | Solarize | Rotation |

stage. To mitigate the disparities between pre-training and fine-tuning for VIS, we adopt a two-pronged strategy called *video per-training*. Firstly, we introduce consistent pseudo-video augmentations at the data level during pre-training to create consistent pseudo-videos from image data used for pre-training. Besides, we propose multi-scale temporal modules for both pre-training and fine-tuning stages. The overall pipeline of *video pre-training* is illustrated in figure 7. In the following, we will introduce consistent pseudo-video augmentations and multi-scale temporal modules.

## 3.1 CONSISTENT PSEUDO-VIDEO AUGMENTATIONS

To create pseudo-videos from the image dataset, we propose consistent pseudo-video augmentations include an auto augmentation technique tailored for videos and, subsequently, a copy & paste augmentation capable of preserving consistency within individual frames and the entire video sequence.

### 3.1.1 VIDEO AUTO AUGMENTATION

The auto augmentation technique, proposed in AugSeg (Zhao et al., 2022), has been found beneficial in the semi-supervised domain due to its capability to search for the optimal augmentation strategy tailored to specific downstream tasks. Our approach has been adapted into a video-level automatic augmentation technique.

Similar to AugSeg, we randomly sample a less than $k$[1] number of augmentations from the augmentation pool, as shown in Tab. 1. However, what differentiates us from AugSeg is that we apply the same augmentations to the entire video sequence. This is done to ensure no abrupt changes between consecutive frames. As depicted in the Tab. 2, extensive rotations for individual frames can introduce dramatic changes between consecutive frames, adversely affecting the video's performance. Thus, we opt for rotations within the range of [15, -15] degrees for pseudo videos, simulating the slow and subtle movements typically observed in most real video instances. As shown in Tab. 3,

---
[1]$k$ is empirically set to 3 in our approach.

applying different augmentations to each frame individually from the augmentation pool (*Image*) can result in significant variations between consecutive frames, ultimately leading to a decrease in overall performance when compared with the setting of applying the same intensity augmentations and similar rotations on all frames in a video.

Video tasks generally have fewer and more challenging annotations compared to image tasks. Therefore, this augmentation strategy tailored for video tasks increases the quantity and diversity of video data by generating pseudo videos.

### 3.1.2 VIDEO COPY & PASTE AUGMENTATION

Most existing augmentation methods such as Copy & Paste or CutMix-based approaches(Ghiasi et al., 2021; Ying et al., 2023; Yun et al., 2019b; French et al., 2019; Yun et al., 2019a) have yielded promising results in image or video segmentation tasks. However, it is often challenging to directly apply image-based augmentation methods to each video sequence frame while maintaining consistency between individual and consecutive frames. As shown in Tab. 3, pasting different instances in each frame can disrupt the video's consistency, leading to adverse effects (the *Image* setting).

Through a thorough analysis of video data, we observed that instances appearing simultaneously in videos often belong to the same class. They even exhibit a high similarity in terms of appearance and size. This phenomenon becomes more pronounced in more complex datasets, which can lead to the problem of ID switching during tracking. However, in most video tasks, instances are often randomly copied from the dataset and pasted into each video frame.

Table 2: Ablation study on using different settings of the rotation angles.

| Rotation angles | $AP^{YV19}$ | $AP^{OVIS}$ |
|:---:|:---:|:---:|
| $\pm 15^o$ | **59.7** | **27.4** |
| $\pm 45^o$ | 57.8 | 26.4 |
| $\pm 60^o$ | 58.8 | 27.2 |

Those existing augmentation approaches can appear abrupt and do not seamlessly integrate with the frames it's pasted onto so the consistency of individual frames cannot be guaranteed. Therefore, to ensure the consistency of both individual frames and the entire video sequence, we propose a novel method called video copy & paste augmentation.

Those existing augmentation approaches can appear abrupt and do not seamlessly integrate with the frames it has pasted onto, so the consistency of individual frames cannot be guaranteed. Therefore, to ensure the consistency of individual frames and the entire video sequence, we propose a novel method called video copy & paste augmentation.

First, we take an image, denoted as $I$, and replicate it $T$ times. Each image undergoes a simple yet essential augmentation process, which in-

Table 3: Ablation experiments of performance comparison between different augmentations on different frames (denoted as *Image*) and the same augmentation on all frames in a pseudo video (denoted as *Video*).

| Augmentations | YTVIS19 | | OVIS | |
|:---:|:---:|:---:|:---:|:---:|
| | *Image* | *Video* | *Image* | *Video* |
| Auto Augmentation | 59.5 | 60.0 | 26.8 | 27.4 |
| Copy &Paste | 59.1 | 59.8 | 26.8 | 27.6 |

cludes random angle rotations, to construct the most basic form of pseudo-video, denoted as $V$. Subsequently, we randomly select one instance from $I$ as a pivotal instance, denoted as $i$, along with its corresponding mask annotation, denoted as $m$. We then mask out all regions in $I$ except, for instance $i$, resulting in $i^I = I \times m$. Following this, we apply minor augmentations, such as flips, scaling, rotations, and coordinate transformations of the instance center, to $i^I$ and $m$ for $T$ iterations, denoted as $A_t$ for $t$-th frame. Finally, we sequentially paste the $T$ frames of $i^I$ into the pseudo-video $V$. Here, each frame $V_t$ in the pseudo-video can be represented as:

$$V_t = V_t \times (1 - A_t(m)) + A_t(i^I) \tag{1}$$

If one instance (its mask $\hat{m}$) is occluded by the pasted instance in $t$-th frame, the mask $\hat{m}_t$ of the occluded instance will be represented as:

$$\hat{m}_t = \hat{m} \times (1 - A_t(m)) \tag{2}$$

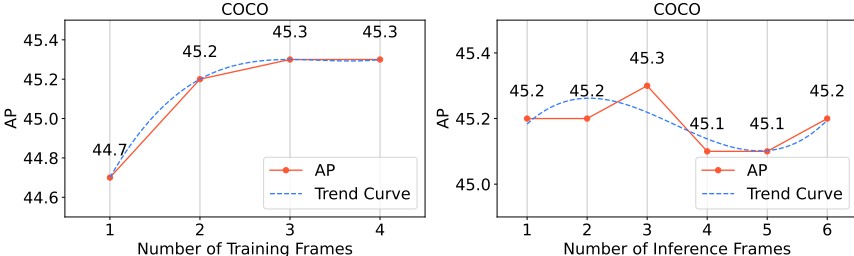

Figure 3: The left and right figures respectively depict the performance with varying frame counts during pre-training, both in the training and inference stages. the optimal performance is achieved when both the training and inference phases generate a three-frame pseudo video.

## 3.2 Multi-Scale Temporal Module

The existing VIS methods introduce temporal modules directly during fine-tuning, rather than during pre-training. However, the current availability of video datasets is limited to only a few hundred thousand videos. This implies a severe lack of diversity in terms of categories and scenes. In such circumstances, direct pre-training on video datasets is infeasible. Therefore, after obtaining pseudo videos in the previous section, we further reduce the gap between pre-training and fine-tuning on videos by introducing temporal modelling during pre-training.

Existing research, such as Mask2Former(Cheng et al., 2022) or Mask DINO(Li et al.), typically employs a multi-scale deformable attention Transformer to extract continuous multi-scale features denoted as $f_s$ ($s$ denotes a scale.). Then, each $f_s$ are input to the multi-scale temporal module for each scale. Specifically, for each scale $s$, we split $f_s \in \mathbb{R}^{T \times H \times W \times D}$ along the temporal dimension into two dense feature sets, $f_s^1$, and $f_s^2 \in \mathbb{R}^{\frac{T}{2} \times H \times W \times D}$, where $T, H, W$, and $D$ represent the number of frames, height, width, and feature dimension, respectively. The

Table 4: Ablation experiments of performance brought by different consistent pseudo-video augmentation techniques in the video pre-training phase. VCP: Video Copy & Paste; VAA: video auto augmentations.

| Rotation | VCP | VAA | AP | $AP^s$ | $AP^m$ | $AP^l$ |
|----------|-----|-----|------|------|------|------|
| ✓ | | | 44.7 | 24.7 | 48.3 | 64.9 |
| ✓ | ✓ | | 44.8 | 24.4 | 48.8 | 65.9 |
| ✓ | | ✓ | 45.0 | 25.3 | 48.7 | 66.5 |
| ✓ | ✓ | ✓ | 45.3 | 25.4 | 49.1 | 66.2 |

split features are then recombined into $F_s^1 = concat(f_s^1, f_s^2)$ and $F_s^2 = concat(f_s^2, f_s^1) \in \mathbb{R}^{T \times H \times W \times D}$. The features $F_s^1$ and $F_s^2$ are then fed into a Swin(Liu et al., 2021) block with self- and cross-attention, and FFN (Feed-Forward Network). During self-attention, $F_s^1$ is used for queries, keys, and values, while during cross-attention, $F_s^1$ serves as queries and $F_s^2$ as keys and values. This process is iteratively performed for $L$ layers in a short-term temporal manner, where the updated $f_s$ from each layer serves as the input for the next. This manner effectively establishes short-term temporal $T_l$ interactions between consecutive frames. Subsequently, we employ the resulting output $f_s^u$ from the Swin block as input for the $N$ ConvGRU (Lin et al., 2022) layers, facilitating the implementation of a long-term temporal interaction manner that spans the entirety of the video sequence. Ultimately, we combine the unaltered feature $f_s$ with the feature after ConvGRU to produce the final output, which can be represented as:

$$f_s^N = ConvGRU(f_s^u) + f_s \qquad (3)$$

The output multi-scale features from the multi-scale temporal module are then input to an FPN block. The output of the FPN block is input to the decoders of Mask2Former to make instance segmentation predictions. The learning objective is the same as Mask2Former.

## 3.3 Datasets

We conducted our experiments and reported our results on three distinct VIS datasets, namely YouTube-VIS 2019(Yang et al., 2019b), YouTube-VIS 2021(Yang et al., 2019b), and OVIS(Qi et al., 2022). More details of these datasets can be found in A.1. These datasets collectively provide a diverse range of challenges, including different video durations, object occlusions, complex motions, and more. They serve as comprehensive benchmarks for assessing the efficacy and robustness of VIS methods. The unique characteristics of each dataset contribute to the holistic evaluation of algorithms in various real-world scenarios, offering valuable insights into the strengths and

| | Methods | YTVIS19 | | | | | YTVIS21 | | | | | OVIS | | | | |
|---|---|---|---|---|---|---|---|---|---|---|---|---|---|---|---|---|
| | | AP | AP$_{50}$ | AP$_{75}$ | AR$_1$ | AR$_{10}$ | AP | AP$_{50}$ | AP$_{75}$ | AR$_1$ | AR$_{10}$ | AP | AP$_{50}$ | AP$_{75}$ | AR$_1$ | AR$_{10}$ |
| ResNet-50 | MaskTrack R-CNN | 30.3 | 51.1 | 32.6 | 31 | 35.5 | 28.6 | 48.9 | 29.6 | 26.5 | 33.8 | 10.8 | 25.3 | 8.5 | 7.9 | 14.9 |
| | Mask2Former-VIS | 46.4 | 68 | 50 | - | - | 40.6 | 60.9 | 41.8 | - | - | 17.3 | 37.3 | 15.1 | 10.5 | 23.5 |
| | SeqFormer | 47.4 | 69.8 | 51.8 | 45.5 | 54.8 | 40.5 | 62.4 | 43.7 | 36.1 | 48.1 | 15.1 | 31.9 | 13.8 | 10.4 | 27.1 |
| | MinVIS | 47.4 | 69 | 52.1 | 45.7 | 55.7 | 44.2 | 66 | 48.1 | 39.2 | 51.7 | 25 | 45.5 | 24 | 13.9 | 29.7 |
| | IDOL | 49.5 | 74 | 52.9 | 47.7 | 58.7 | 43.9 | 68 | 49.6 | 38 | 50.9 | 30.2 | 51.3 | 30 | 15 | 37.5 |
| | VITA | 49.8 | 72.6 | 54.5 | 49.4 | 61 | 45.7 | 67.4 | 49.5 | 40.9 | 53.6 | 19.6 | 41.2 | 17.4 | 11.7 | 26 |
| | GenVIS | 51.3 | 72.0 | 57.8 | 49.5 | 60.0 | 46.3 | 67.0 | 50.2 | 40.6 | 53.2 | 35.8 | 60.8 | 36.2 | 16.3 | 39.6 |
| | DVIS | 52.6 | 76.5 | 58.2 | 47.4 | 60.4 | 47.4 | 71.0 | 51.6 | 39.9 | 55.2 | 33.8 | 60.4 | 33.5 | 15.3 | 39.5 |
| | CTVIS | 55.1 | 78.2 | 59.1 | **51.9** | 63.2 | 50.1 | 73.7 | 54.7 | 41.8 | 59.5 | 35.5 | 60.8 | 34.9 | 16.1 | 41.9 |
| | **Ours** | **56.0** | **78.6** | **60.8** | 51.3 | **63.4** | **53.4** | **75.9** | **57.6** | **45.2** | **61.4** | **39.5** | **65.4** | **39.1** | **17.4** | **45.2** |
| Swin-L | SeqFormer | 59.3 | 82.1 | 66.4 | 51.7 | 64.6 | 51.8 | 74.6 | 58.2 | 42.8 | 58.1 | - | - | - | - | - |
| | Mask2Former-VIS | 60.4 | 84.4 | 67 | - | - | 52.6 | 76.4 | 57.2 | - | - | 25.8 | 46.5 | 24.4 | 13.7 | 32.2 |
| | MinVIS | 61.6 | 83.3 | 68.6 | 54.8 | 66.6 | 55.3 | 76.6 | 62 | 45.9 | 60.8 | 39.4 | 61.5 | 41.3 | 18.1 | 43.3 |
| | VITA | 63 | 86.9 | 67.9 | 56.3 | 68.1 | 57.5 | 80.6 | 61 | 47.7 | 62.6 | 27.7 | 51.9 | 24.9 | 14.9 | 33 |
| | IDOL | 64.3 | 87.5 | 71 | 55.5 | 69.1 | 56.1 | 80.8 | 63.5 | 45 | 60.1 | 42.6 | 65.7 | 45.2 | 17.9 | 49.6 |
| | GenVIS | 64.0 | 84.9 | 68.3 | 56.1 | 69.4 | 59.6 | 80.9 | 65.8 | 48.7 | 65.0 | 45.4 | 69.2 | 47.8 | 18.9 | 49.0 |
| | DVIS | 64.9 | **88.0** | **72.7** | **56.5** | 70.3 | 60.1 | 83.0 | 68.4 | 47.7 | 65.7 | 48.6 | **74.7** | 50.5 | 18.8 | 53.8 |
| | CTVIS | **65.6** | 87.7 | 72.2 | **56.5** | **70.4** | 61.2 | **84** | 68.8 | 48 | 65.8 | 46.9 | 71.5 | 47.5 | 19.1 | 52.1 |
| | **Ours** | 65.1 | 86.0 | 71.7 | 56.1 | 69.9 | **62.2** | 83.1 | **69.1** | **48.8** | **67.1** | **49.4** | 72.9 | **52.5** | **20.1** | **54.2** |

Table 5: Quantitative comparison between our method with previous state-of-the-art methods. The best and second best are highlighted by **bold** and underlined numbers, respectively.

weaknesses of different approaches. Besides, the pseudo-videos are constructed on the widely used instance segmentation dataset, COCO (Lin et al., 2014).

# 4 EXPERIMENTS

## 4.1 IMPLEMENTATION DETAILS

**Model Setting.** We adopt the framework of instance segmentation methods such as Mask2Former or Mask DINO as our instance segmentation network. Unless otherwise specified, ResNet-50(He et al., 2016) is set as the backbone. To ensure a fair comparison with state-of-the-art methods, we maintain all the original settings of the baseline methods(Cheng et al., 2022; Ying et al., 2023) without any modifications. For the sake of a fair comparison, we have exclusively compared other methods with DVIS when it performed inference with a 480px setting on OVIS benchmark.

**Data Augmentation.** We employ the proposed consistent pseudo-video augmentation technique to create pseudo videos from images in the COCO dataset for the pre-training phase. During the fine-tuning phase, the pseudo-videos and VIS datasets are used to train VIS models jointly. During pre-training and fine-tuning, the pseudo video sequences consist of 3 frames and 10 frames, respectively.

## 4.2 MAIN RESULTS

We conducted a comparative analysis of our approach using both ResNet-50 and Swin-L as backbones against SOTA methods on the three primary VIS benchmarks: YouTubeVIS 2019, YouTubeVIS 2021, and OVIS. The performance and visualization results are presented in Tab. 5 and A.5, respectively.

**YouTubeVIS 2019** predominantly consists of rather simplistic short videos. Consequently, with a ResNet-50 backbone, our method on YouTubeVIS 2019 showed a mere 0.9% AP increase. Interestingly, leveraging the Swin-L backbone in our method on YouTubeVIS 2019, our approach lagged the SOTA by 0.5% AP. This disparity can be attributed to the lack of complexity in the videos, rendering them less sensitive to temporal modules.

**YouTubeVIS 2021**, an extension of YouTubeVIS 2019, includes a broader array of longer and more intricate videos. In this context, our approach exhibited noteworthy improvements, yielding a 3.3% AP boost with ResNet-50 as the backbone and a 1.0% AP enhancement with Swin-L as the backbone.

**OVIS**, featuring the longest and most intricate video sequences among the three benchmarks, serves as a litmus test for evaluating the effectiveness of video pretraining. Our approach demonstrated significant performance gains, with a 4.0% AP improvement using ResNet-50 as the backbone and a 2.5% AP enhancement with Swin-L as the backbone. This underscores the notion that our video pretraining method excels particularly in more complex video benchmarks.

## 4.3 ABLATION STUDIES

Given the absence of annotated validation sets in the benchmarks for VIS tasks, we carefully partitioned the labeled training sets (YouTube2019 and OVIS) into custom training splits and validation

splits, identical to the Wu et al. (2022b). These splits were instrumental in our efforts to conduct rigorous ablation experiments. Furthermore, it's worth noting that all experiments related to VIS, were meticulously conducted using these custom split datasets to ensure the robustness and credibility of our findings.

**Pre-training.** In Tab. 4, we initially validate the effectiveness of various video data augmentation methods for generating pseudo-videos. Using rotation alone as a data augmentation method yields a 1.0% AP improvement over the instance segmentation baseline on COCO benchmark. Furthermore, adding video copy & paste and video auto augmentation on top of rotation results in additional improvements of 0.1% and 0.3% AP, respectively. When all augmentations are applied to the pseudo-video inputs during pre-training, it leads to a substantial 1.6% AP improvement over the baseline.

In Tab. 6, without using data augmentation, we observe that incorporating either the long-term or short-term temporal modules on top of the baseline improves performance by 0.9% and 0.8% AP, respectively. When the complete multi-scale temporal module is used, our method achieves a 1.0% AP improvement.

Moreover, we notice that generating three frames pseudo video as input during pre-training yields the best performance, outperforming single-frame inputs by 0.6% AP. Increasing the number of frames in the pseudo-videos during pre-training does not lead to further improvements. During the inference phase, we find that copying the input image the same number of times as the pseudo-video frames results in the

Table 6: Performance comparison of different temporal modules in the video pre-training phase.

| Long-term | Short-term | AP | $AP_s$ | $AP_m$ | $AP_l$ |
|-----------|-----------|------|------|------|------|
| | | 43.7 | 23.4 | 47.2 | 64.8 |
| ✓ | | 44.6 | 24.3 | 48.5 | 65.5 |
| | ✓ | 44.5 | 24.4 | 48.3 | 64.8 |
| ✓ | ✓ | 44.7 | 24.7 | 48.3 | 64.9 |

best performance, with a 0.2% AP difference between the best and worst performances. Our approach can be easily transferred to other instance segmentation methods, yielding notable performance gains, as shown in A.3.

**Fine-tuning** We choose CTVIS as the baseline for our method since it currently represents the state-of-the-art approach. However, it's important to note that not all instance segmentation methods are suitable as baselines for the pre-training phase of VIS tasks, shown in A.3, Mask2Former proves to be the most suitable choice.

Tab. 7 provides a clear illustration of the impact of video pre-training(VP) when compared to image pre-training alone. Using the same multi-scale temporal module(MSTM) module, we observed an improvement of 0.7% AP and 3.7% AP in the two benchmarks, respectively, when video pre-training was employed. However, when all the proposed modules included: MSTM and consistent pseudo-video augmentations(VA) were used in conjunction and fine-tuned on top of video pre-training, the performance gains were even more significant, with enhancements of 1.6% AP and 4.1% AP in the two benchmarks, respectively.

Table 7: Ablation study on using different settings of the proposed method. The basic setting is image pre-training and video finetuning. CPVA: consistent pseudo-video augmentations; VP: video pre-training.

| MSTM | CPVA | VP | $AP^{YV19}$ | $AP^{OVIS}$ |
|------|------|------|------|------|
| | | | 59.7 | 27.4 |
| ✓ | | | 59.8 | 28.3 |
| ✓ | ✓ | | 60.2 | 28.5 |
| ✓ | | ✓ | 60.5 | 32.0 |
| ✓ | ✓ | ✓ | 61.8 | 32.6 |

## 5 Conclusion

The often-overlooked disparity between image pre-training and video fine-tuning has prompted our novel approach to video pre-training. Its primary objective is to bridge the gap between these two stages and elevate the performance on VIS benchmarks. At the data level, we employ consistent pseudo-video augmentations to construct pseudo-videos that simulate real-world video scenarios. At the modelling level, we introduce a multi-scale temporal module during pre-training. This augmentation strategy yields improvements in both instance segmentation and VIS benchmark performance, with more significant gains observed in increasingly complex VIS benchmarks.

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

# A  APPENDIX

## A.1  DATASETS

- **YouTube-VIS 2019** serves as the inaugural and largest dataset designed explicitly for VIS. It encompasses 2,238 training videos, 302 videos for validation, and 343 test videos, all characterized by high resolution. The average duration of these YouTube video clips is approximately 4.61 seconds. This dataset comprises 40 object categories.
- **YouTube-VIS 2021** is an extension and enhancement of the YouTube-VIS 2019 dataset. It retains the same number of object categories, i.e., 40, but features slight variations in the category label set. The dataset expands to include 2,985 training videos and 453 validation videos. The primary focus of this extension is the improvement in instance annotation quality.
- **OVIS (Occluded VIS)** is a relatively recent and challenging addition to the VIS dataset landscape. It consists of 607 training videos, 140 validation videos, and 154 test videos. Notably, OVIS videos tend to be significantly longer, with an average duration of approximately 12.77 seconds. More importantly, this dataset contains videos that record instances with severe occlusion, complex motion patterns, and rapid deformations, making it an ideal benchmark to evaluate and analyze the performance of various methods.

| | Methods | Params. | Epochs | Query type | AP | $AP_{50}$ | $AP_{75}$ | $AP_s$ | $AP_m$ | $AP_l$ |
|---|---|---|---|---|---|---|---|---|---|---|
| ResNet-50 | Mask2Former | 44M | 50 | 100 | 43.7 | 66.0 | 46.9 | 23.4 | 47.2 | 64.8 |
| | Mask DINO | 52M | 50 | 300 | 46.3 | 69.0 | 50.7 | 26.1 | 49.3 | 66.1 |
| | Our + Mask2Former | 54M | 50 | 100 | 45.3 | 68.3 | 48.9 | 25.4 | 49.1 | 66.2 |
| | Our + MaskDINO | 57M | 50 | 300 | 47.2 | 70.2 | 51.8 | 27.5 | 50.5 | 67.3 |
| Swin-L | Mask2Former | 216M | 100 | 200 | 50.1 | - | - | 29.9 | 53.9 | 72.1 |
| | Our + Mask2Former | 227M | 50 | 200 | 50.5 | 74.9 | 54.9 | 29.6 | 54.6 | 72.8 |

Table 9: Ablation experiments of performance brought by image pre-training and video pre-training, we conducted a comparative analysis between two distinct instance segmentation methods, both utilizing the ResNet-50 backbone.

## A.2  PRE-TRAINING INFERENCE OF OURS

Since the input is transformed into a pseudo-video consisting of T frames, during the final inference phase, we replicate the same image T times to simulate the input for the video. Ultimately, the output at the last frame is used as the output for the inference stage. As depicted in figure 3, when T identical images enter the temporal module, it is akin to iteratively refining the same image T times. Clearly, the performance is optimized when the number of replications matches the number of refinements.

Table 8: Ablation experiments of performance brought by different video pre-trained models.

| Video Pre-trained Method | $AP^{YV19}$ |
|---|---|
| Mask DINO | 60.5 |
| Mask2Former | 61.8 |

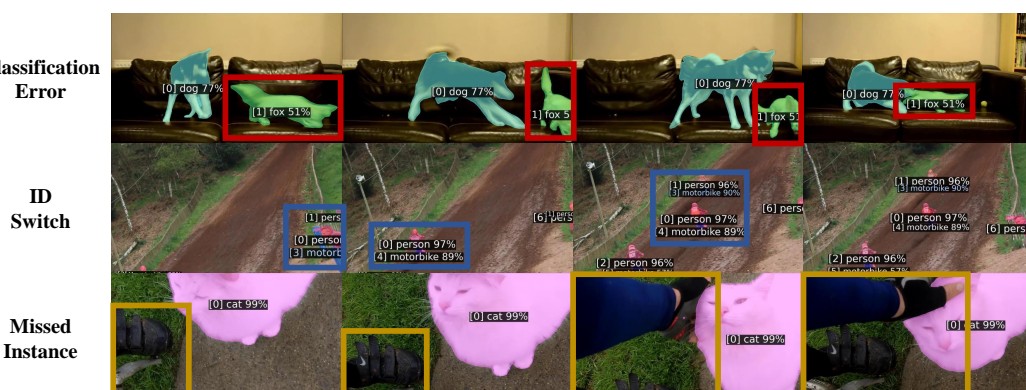

Figure 4: Failure cases

### A.3 PRE-TRAINING COMPARISON

Our approach can be easily transferred to other instance segmentation methods, yielding notable performance gains. For example, as shown in Tab. 8, when using ResNet-50(He et al., 2016) as the backbone, our method achieves a 1.6 AP% improvement over Mask2Former and a 0.9% AP improvement over Mask DINO. When Swin-Large serves as the backbone, our method outperforms Mask2Former by 0.4% AP with only 50 training epochs.

Despite Mask DINO surpassing Mask2Former by 1.9% AP during the pretraining phase, as shown in Tab. 9, our method fine-tuned on Mask2Former outperforms our method fine-tuned on Mask DINO-based approach by 1.3% AP under the YouTubeVIS 2019 benchmark. Therefore, for all subsequent ablation experiments in VIS, we adopted the same pre-training baseline as CTVIS, which is Mask2Former.

Table 10: Ablation study on different benchmarks of the FPS.

| ResNet-50 | YV19 | OVIS |
|-----------|------|------|
| FPS | 9.7 | 2.9 |

### A.4 FAILURE CASES

As shown in figure 4, when using the ResNet-50 backbone, frequent occurrences of category detection errors, ID switches, and instances going undetected are observed. These issues present promising avenues for future research and improvement.

### A.5 VISUALIZATION RESULTS

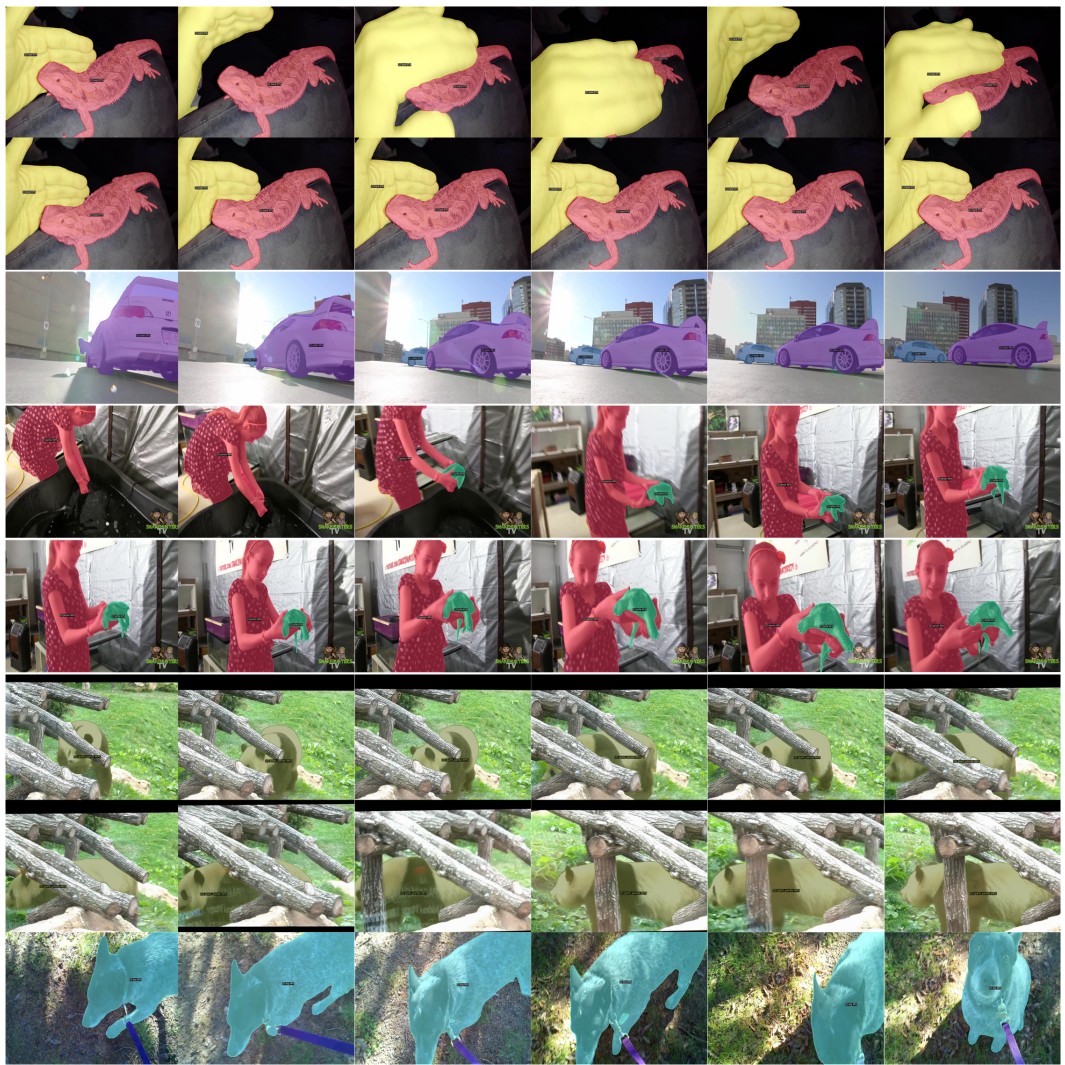

Figure 5: YouTubeVIS 2019 Visualization Results

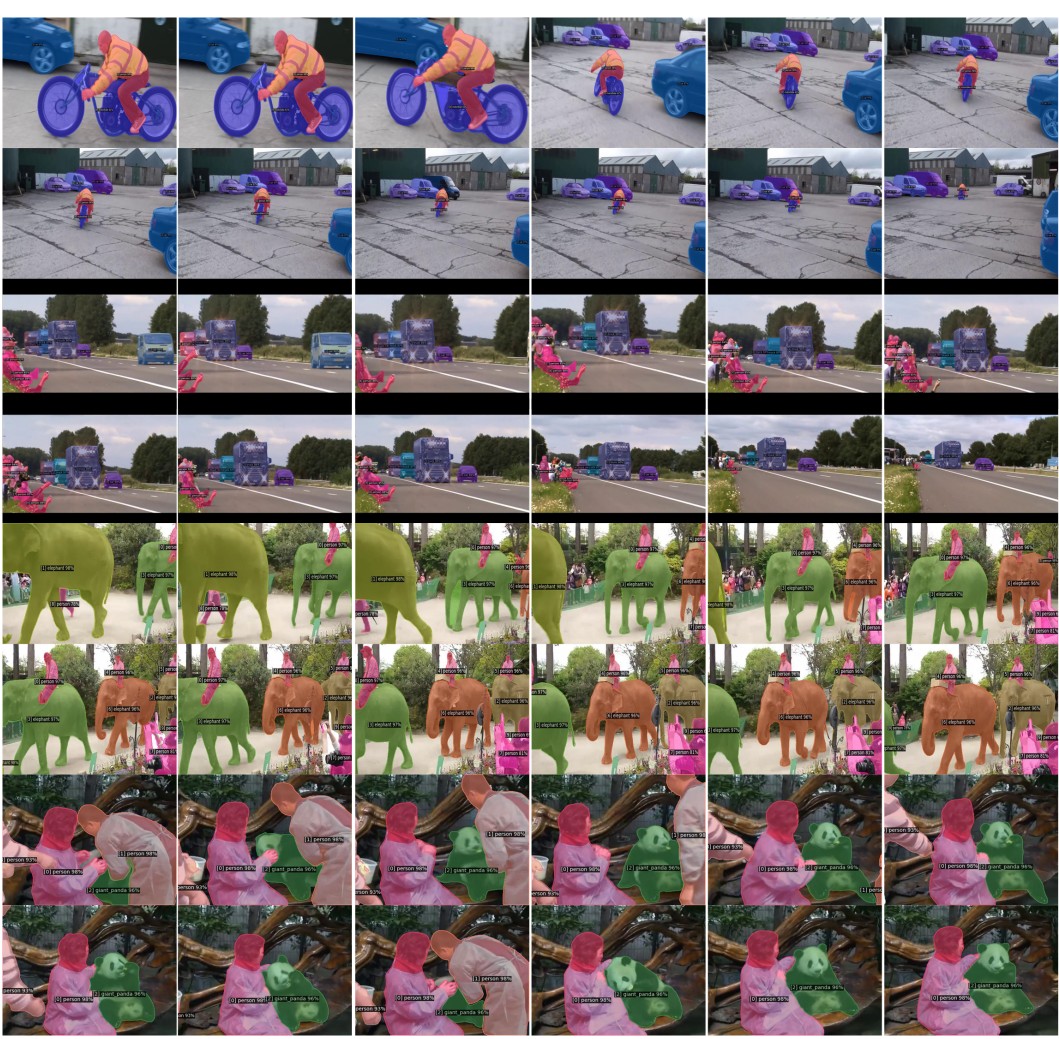

Figure 6: YouTubeVIS 2021 Visualization Results

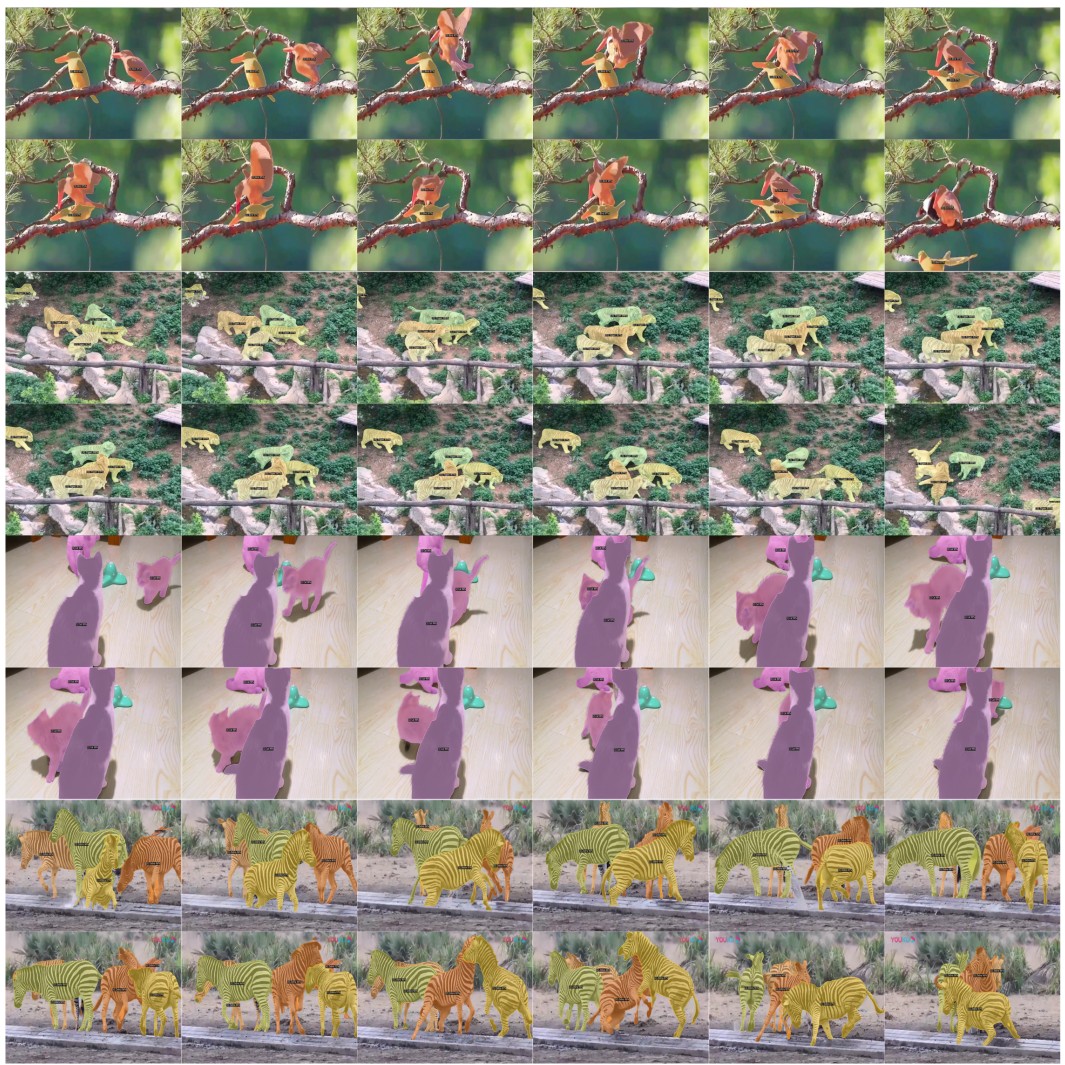

Figure 7: OVIS Visualization Results

