# OpenReview forum: "Amortising the Gap between Pre-training and Fine-tuning for Video Instance Segmentation"
_ICLR.cc/2024/Conference — ICLR 2024 Conference Withdrawn Submission_

### Official Review · Reviewer_aJG2 · 2023-10-25

**Soundness:** 2 fair
**Presentation:** 2 fair
**Contribution:** 2 fair
**Rating:** 5
**Confidence:** 4

**Summary:**

This paper introduces a video pretraining method to enhance VIS (Video Instance Segmentation) performance and reduce the gap between pretraining and fine-tuning stages at the data and modeling levels. Specifically, in terms of data augmentation, this work introduces consistent pseudo video augmentation to maintain instance prediction consistency between the two stages. Furthermore, on the modeling front, it combines multi-scale temporal modules to enhance the model's understanding of time. In three VIS benchmark tests, the proposed approach in this paper outperforms all state-of-the-art methods.

**Strengths:**

1. The motivation behind addressing the gap between pretraining and fine-tuning in this paper is well-founded because simply transferring image pretraining to video fine-tuning does not effectively solve VIS (Visual Inertial Sound) problems.
2. The quantitative results of this work are impressive, surpassing previous state-of-the-art methods, as shown in Table 3 of the paper.

**Weaknesses:**

1. The video data augmentation employed in this work is not particularly novel. It primarily applies existing image adaptation methods to address the issue of video data augmentation. There are already well-established pseudo-video data augmentation methods in the field of VIS, such as [a], which also use techniques like rotation, cropping, and copy & paste for data augmentation. The paper does not provide a detailed explanation in Section 2.3 of how their approach differs from these existing methods. The paper presents a strategy for generating single-frame pseudo-video frames in Eq. (1), but it does not thoroughly explain how to generate continuous videos with natural and smooth transitions between pseudo-video frames. [a] Ctvis: Consistent training for online video instance segmentation. In IEEE Int. Conf. Comput. Vis., 2023.
2. Another module in this work is the multi-scale temporal module (MSTM), which is a combination of Swin block (Liu et al., 2021) and ConvGRU (Lin et al., 2022), with the input being a simple concatenation of pseudo-video frames. Self-attention, cross-attention, and ConvGRU are relatively common components, and the paper should provide a detailed explanation and analysis of the role of this MSTM.
3. The experimental results in this paper are not well-explained. For example, it is unclear which datasets and parameter configurations were used to generate the results in Table 2, Table 3, Table 4, and Table 6. In particular, in the last row of Table 7, the results for AP(YV19) and AP(OVIS) do not seem to correspond to the performance results in Table 5.

**Questions:**

There are some minor issues with details in this paper:
1. What is the relationship between Eq. (2) and Eq. (1)? How is Eq. (2) used to generate pseudo-video frames?
2. The presentation in this paper has significant issues. There is a lot of repetition throughout the paper. For example, the passages "Given the scarcity of video data..." and "Many online VIS..." are repeated in both page 2 and page 3. The content "Those existing augmentation approaches can appear..." is duplicated on page 6. In the caption of Figure 3, "the optimal performance is" should be corrected to "The optimal performance is achieved." The section "4.3 Ablation studies" on page 8 is placed incorrectly.

---

### Official Review · Reviewer_RhiN · 2023-10-28

**Soundness:** 2 fair
**Presentation:** 1 poor
**Contribution:** 1 poor
**Rating:** 3
**Confidence:** 5

**Summary:**

This paper focuses on a key problem for VIS task, i.e., the disparities between the pre-training and fine-tuning stages, from the data and model aspects. The authors proposed a consistent pseudo-video augmentation solution to maximize the consistency among the pseudo-videos. The authors test the proposed model on three VIS datasets, youtube-vis 2019/2021, ovis.

**Strengths:**

1.A new data augmentation solution is proposed for generating high-quality pseudo-videos. This is a notable merit.

2.Also, a multiple-scale strategy is proposed during the pre-training stage, which brings performance advantages.

**Weaknesses:**

1.For writing, many repeated paragraphs in the introduction section and sec. 3.1.2.  Also, sec 4.3 can not be read.

2.Novelty is a big issue. In the method section, the proposed data augmentation comes from Augseg while multi-scale temporal module comes from the multi-scale deformable attention Transformer. From this view, the contribution is combining many exsiting techniques rather than proposing a new one.

3.  In the experimental section, the experimental discussion is weak for each part.  Also, the complexity of the whole method should be clarified.

4. For swin-L backbone, the proposed strategy has no performance promotion when compared to the counterparts.

**Questions:**

See the weakness mentioned above.

---

### Official Review · Reviewer_7Ge9 · 2023-10-29

**Soundness:** 2 fair
**Presentation:** 1 poor
**Contribution:** 1 poor
**Rating:** 3
**Confidence:** 5

**Summary:**

With growing interests in the video segmentation tasks, the lack of video annotations has been a bottleneck to improving the accuracy of video models. This paper aims to alleviate such problems by suggesting new augmentation methods. Leveraging the proposed augmentations, this paper can narrow down the discrepancy between pre-training and fine-tuning phases. Additionally, this paper adopts ConvGRU and swin-variant method to model long&short-term temporal modeling. Finally, the authors achieve state-of-the-art performance on multiple VIS benchmarks.

**Strengths:**

Gathering video annotation obviously requires tremendous human-labor compared to that of images.
In this regard, there have been a number of approaches to alleviate the lack of video datasets.
Therefore, the paper's motivation is straight-forward and the authors achieve state-of-the-art results on multiple benchmarks.

**Weaknesses:**

Practicality:
- It is somewhat obvious that involving video-like pre-training would improve the accuracy, and it is more like an engineering aspect.
- One of the main reasons why previous works couldn't apply pseudo-video training at the pre-training phase would be the computing resource issue. I believe this approach should at least reduce the computation during the fine-tuning stage to validate its effectiveness.
- It is unclear if this method is benefitted from the "temporal" pre-training. Specifically, the augmentations that this paper uses also help image segmentation capacity. For instance, Table 4 and 6 show that image instance segmentation accuracy improves. However, as this paper is claiming that such augmentations "temporal" modeling, it is ambiguous if the improvements in the VIS benchmarks are driven from temporal modeling or massive computation involved during training.
- This paper uses video sequence of 10 frames, which would require a gigantic GPU memory.

Methodology:
- I do not understand why the philosophy behind splitting features and recombining them. What's the necessity of this?
- The method is supposed to be modeling "temporal" characteristics. Especially, because of the use of ConvGRU which models temporal aspects in a small sized conv window, pseudo videos must be somewhat temporally consistent. For instance, such videos should model natural movements of objects. However, the presented augmentation is mostly conducted with randomness, not fully considering the natural video aspects.

Lack of supporting experiments:
- Just showing the improvements in terms of accuracy cannot validate the effectiveness of this augmentation as aforementioned.
- What specific temporal modeling property does this augmentation help?

Poor writing quality:
- So many duplicates of sentences and paragraphs.
    - Intro paragraphs 3-4 and 5-6.
    - Paragraph 3 and 4 of Section 3.1.2.
    - Many other sentences that essentially say the same thing.
- I believe equation 1, 2, 3 are not necessary.
- Not clear which model they are using. "Model Setting" part does not explain it in detail.
- Typos and errors:
    - Page 5 - "The overall pipeline of video pre-training is illustrated in figure 7" : not figure 7.
    - Many other grammatical errors.

**Questions:**

- How much iterations are executed during pre-training and fine-tuning?
- What & how many GPUs are being used?
- Statistical Significance: For this type of paper, I strongly believe that the method should deliver stable outputs. How many runs were executed to come up with the scores? Are the numbers a mean/median?

- Listed other questions in the weaknesses section, too.

---

### Official Review · Reviewer_vE9c · 2023-10-30

**Soundness:** 2 fair
**Presentation:** 2 fair
**Contribution:** 2 fair
**Rating:** 3
**Confidence:** 5

**Summary:**

The paper proposes to reduce the gap between pre-training and fine-tuning in existing video instance segmentation methods, where pre-training is  on images while fine-tuning is on videos. Consistent pseudo-video augmentations and multi-scale temporal module are proposed to solve the issue. The experiments are validated on YTVIS19, YTVIS21 and OVIS.

**Strengths:**

1. The paper has a clean idea and motivation, which is easy to understand.

2. Table 7 provides a detailed study on the proposed each component.

**Weaknesses:**

1. The paper has a limited tech novelty, where the proposed pseudo-video augmentation is based on copy & paste and AutoSeg. According to Table 7, the proposed CPVA only brings 0.4 AP improvement on YTVIS19 and 0.2 AP improvement on OVIS, which is negligible.

2. The writing of the paper is not well organized. For example, there are two duplicated paragraphs with the same content in Sec 3.1.2. Also, the margin Sec 4.3 is too small to read.

3. The paper misses a good illustration figure on the proposed pseudo-video augmentations.

4. Can the author provide explanation on the numerical results correspondence between Table 5 and Table 7?

5. Missing related VIS works in the related work section, where [b] also uses pseudo videos to improve performance:
[a] Efficient video instance segmentation via tracklet query and proposal. CVPR, 2022.
[b] Mask-Free Video Instance Segmentation. CVPR, 2023.
[c] Video instance segmentation tracking with a modified vae architecture. CVPR, 2020.

**Questions:**

Can the authors also show the effect of their training strategy on the video panoptic segmentation or multiple object tracking and segmentation benchmarks with complex scene?